# Development of a More Sustainable Hybrid Process for Lithium and Cobalt Recovery from Lithium-Ion Batteries

**José Cristiano Mengue Model \*** and **Hugo Marcelo Veit**

Graduate Program in Mining, Metallurgy and Materials Engineering, Laboratory of Corrosion, Protection and Materials Recycling, Federal University of Rio Grande do Sul, Av. Bento Gonçalves, 9500, 43426 Building, Agronomia, Porto Alegre 91501-970, RS, Brazil; hugo.veit@ufrgs.br
**\*** Correspondence: jose.model.jc@gmail.com; Tel.: +55-5133-089-432

**Abstract:** Lithium-ion batteries are widely used as a power source for portable devices and electrical vehicles (EVs). After their useful life, they can provide a secondary source from which to obtain some materials which make them up, such as lithium and cobalt. However, the metallurgical route which will be used to recover them must be considered. Therefore, is crucial that many efforts to develop more environmentally favorable recovery processes be pursued. Due to this, the present work aimed to use 1.5 M DL-malic acid and compare it to 2 M sulfuric acid, employing heat pretreatment of 1 h and 3 h to remove the powder cathode binder polyvinylidene fluoride (PVDF); for all conditions, experiments were carried out with and without adding the oxidizing agent hydrogen peroxide. The PVDF temperature degradation occurred at 630 °C. The best yields occurred in the presence of $H_2O_2$ 10% *v/v* and heat pretreatment. With sulfuric acid (1 h) it was possible to recover 33.49% Co and 4.63% Li, and (3 h) 36.36% Co and 4.64% Li. With DL-malic acid it was possible to recover (1 h) 29.78% Co and 3.44% Li, and (3 h) 32.73% Co and 3.99% Li.

**Keywords:** recycling; DL-malic acid; lithium-ion battery; eco-friendly; pretreatment





## 1. Introduction

Lithium-ion batteries (LIBs) are widely used in different kinds of technology as an energy storage device, such as handheld devices (e.g., smartphones and tablets) or electric vehicles (EVs) [1]. Their wide applicability is due to their superior electrical performance, such as high energy density, long life cycle, and no memory effect, and their lighter weight when compared to other types of batteries (e.g., lead–acid, nickel–metal hydride, and nickel–cadmium batteries) [2]. The LIB concept was proposed by different researchers in the 1970s [3]. Many innovations have been attributed to Yoshino for the development of rechargeable lithium-ion batteries, who registered the first patent [4].

Lithium-ion batteries are composed of a cathode—transition metal powders such as cobalt, manganese, and nickel are usually used—a graphite anode, a porous polymeric membrane, which can allow the electrons to flow during charge and discharge processes, and an electrolyte—the means by which the flow of electrons occurs. This electron flow is generated by the movement of lithium ions between the cathodic and anodic powders. The cathode and anode collectors are composed of aluminum and copper, respectively. Polyvinylidene fluoride (PVDF) is a polymer used as a binder to adhere cathode (lithium-cobalt oxide) and anode (graphite) powders into support sheets [5,6], which can be a major obstacle to improving the efficiency of hydrometallurgical routes aiming to recover lithium and cobalt for recycling.

The largest lithium producers in 2020 were Australia and Chile, responsible for 48.1% and 26.0% of the world's total lithium production, respectively. Among various industrial applications, 74% of lithium is destined for the manufacture of batteries and 14% for the ceramic and glass industry [7]. Regarding cobalt, the majority of the world's production

in 2020 was attributed to the Democratic Republic of Congo with 68.9%, while Russia and Australia were responsible for 6.3% and 4.0%, respectively [7]. Around 46% of the cobalt produced in 2018 was used in the manufacture of batteries, and according to a report by the German Mineral Resources Agency (DERA) [8], this demand is due to intense EV development.

When LIBs are discarded, alone or together with some equipment (e.g., smartphones, notebooks, tablets, GPS devices, etc.), they are denominated as waste from electrical and electronic equipment (WEEE). In general, the WEEEs contain about 60 metals, such as copper, cobalt, gold, platinum, lithium, silver, palladium, etc. Developing efficient methodologies for recovering these metals, in addition to avoiding the cost of extracting them from ores, would be environmentally favorable by reducing the impact of mining and by reducing pollution due to incorrect destinations given to WEEEs [9,10]. The useful life of LIBs, used as a power source for smartphones, is about 2 years, or 300 to 500 cycles [6,7,11,12], and represents a significant contribution to the generation of WEEEs [5]. In cases of incorrect disposal, after their useful life cycle, LIBs can cause harmful environmental impacts due to the materials they are made of; moreover, they are an important source of raw materials for the recovery of metals with economic added value [13]. Pretreatment processes, hydrometallurgy, and pyrometallurgy are the most extensively studied/employed kinds of recycling processes [14].

There is no single definition of pretreatment, a usual preceding step adopted in hydrometallurgical processes, for recycling LIBs. Some authors [15,16] usually divide it into mechanical separation, mechanical–chemical processes, thermal treatment, and dissolution processes. Pretreatment was divided by Yao et al. [17] into unloading, disassembly, and cathodic material separation, while Zhang et al. [18] described the process as manual treatment, disassembly and classification, comminution (mechanical treatment), sieving, separation by particle size, and mechanical–chemical treatment. Due to it improving the yields recovered, the pretreatment process is widely used as a preparatory step in hydrometallurgy, as it was used in the present work.

Hydrometallurgical processes consist of dissolving metals in acidic or basic leaching solutions to extract them from waste. The most common methodologies use strong inorganic acids as leaching agents (e.g., $HNO_3$, $H_2SO_4$, and $HCl$). These processes are not considered environmentally friendly, as they release vapors and gases (NOx, $SO_3$, and $Cl_2$), and the solutions can permeate the soil when poorly managed or, in cases of accidents, cause damages to water resources and biodiversity, including harms to human beings [19]. Alternative proposals to replace inorganic acids with organic acids, which are less harmful to the environment, have been studied. Musariri et al. used citric acid ($C_6H_8O_7$) and DL-malic acid ($C_4H_6O_5$), both 1.5 M, with the addition of $H_2O_2$ 2% *v/v* as an oxidizing agent and a temperature of 95 °C. Citric acid was the most efficient agent, and it dissolved up to 95% of lithium and cobalt [20]. With an aqueous mixture of citric and ascorbic acid ($C_6H_8O_6$), Nayaka et al. [21] leached obsolete LIBs. Copper and lithium were obtained in the form of cobalt oxalate and lithium fluoride by selective precipitation and the addition of oxalic acid ($C_2H_2O_4$) and ammonium fluoride, respectively.

The presence of $H_2O_2$ in the leaching process for some metals results in valence decreases, such as $Co^{3+}$ becoming the more soluble $Co^{2+}$. Some studies corroborate that the presence of an oxidizing agent improves the performance of metal recovery for both organic and inorganic acids [22]. In their work, Sattar et al. carried out leaching with 3M $H_2SO_4$ at 90 °C and recovered 92% of Li, 68% of Co, and 34.8% of Mn without adding $H_2O_2$. After the addition of 4% *v/v* $H_2O_2$, the metal leaching efficiency increased by more than 98% [23].

In pyrometallurgy, the use of furnaces at high temperatures aims to reduce the oxides to a metallic alloy. Gases and slag also result from this process. In recycling LIBs, the great advantage of performing a pyrometallurgical process is that, in addition to being processed in a single batch, it does not require pretreatment to concentrate the material, as is usually carried out in hydrometallurgy [18,24]. LIB recycling operations, on an industrial

scale, are more commonly conducted via pyrometallurgy. The main plants are located in North America, Europe, and Asia. Umicore has two pyrometallurgical processing plants in Belgium and China with capacities of 7000 t/year and 5000 t/year, respectively, while Retriev has a plant using a hydrometallurgical process in the United States/Canada, with a capacity of 4500 t/year. All pyrometallurgical routes adopt high temperatures, each one according to specific process parameters.

This work aimed to develop a hybrid route, mixing a heat pretreatment step—at around 650 °C, not so high as pyrometallurgy because it is intended only to decompose the PVDF binder—and a following hydrometallurgical step that should be more environmentally favorable due to the use of 1.5 M DL-malic acid instead of an inorganic acid. A hydrometallurgical process with 2 M sulfuric acid was used as a control. The addition of hydrogen peroxide $H_2O_2$ 10% *v/v* as an oxidizing agent was also evaluated—an optimized condition, according to the work of Dutta et al. [25]. This hybrid condition makes this route innovative, as the vast majority of studies focus either on hydrometallurgy or pyrometallurgy exclusively.

## 2. Materials and Methods

### 2.1. LIB Collection

Batteries were collected from cell phone repair shops. Firstly, 404 LIBs were selected and sorted out from other manufacturing technologies. For the 5 most common brands, a sample was taken from each LIB to characterize the materials that compose it.

### 2.2. Characterization

For the characterization process, LIBs were primarily discharged by short-circuiting them, to eliminate any explosion/ignition risk during the disassembly step. Afterward, the samples were manually opened, and the housing case was removed to proceed with the manual scraping of the cathode and anode powders from their respective collectors. About 1 cm$^2$ each of the housing case and the cathode and anode collectors were separated to be analyzed by XRF—X-ray fluorescence (Thermo Scientific, Niton xl3t model, Waltham, MA, USA). Following that, the digestion of the cathode powder was carried out, assisted by microwaves, according to method 3051A EPA, and subsequently, the contents of elements were quantified by ICP-OES—inductively coupled plasma optical emission spectroscopy (Agilent, 5120 model, Santa Clara, CA, USA). The cathode powder was also analyzed by XRD—X-ray diffraction (Siemens—BRUKER AXS, D-5000 model, Saint Paul, MN, USA)—XRF, and SEM—scanning electron microscopy (Phenon World, PW-100-017 model, Roskilde, Denmark).

A thermal characterization by TGA—thermogravimetric analysis (TA Instruments, model SDT Q600, New Castle, DE, USA)—was carried out on a pure PVDF polymeric sample to obtain the thermal degradation behavior curve of the material to enable the pretreatment step, via combustion, to remove the binder from the cathode and anode powders in the comminuted LIB samples.

### 2.3. Hybrid Processing

2.3.1. Comminution and Granulometric Separation

To crush the batteries (399—the total selected except for the 5 used in the characterization process), a knife mill (Retsch, SM300 model, Haan, Germany) was used. In the first stage, a sieve with an opening of 10 mm was used, and the resulting mass was placed again in the mill to reach a smaller final granulometry; however, this time, a sieve with an opening of 2 mm was used.

After comminution, granulometric separation was performed using a set of sieves (Bertel) with openings of 1 mm and 500 μm to divide the total mass into three fractions according to particle size. The characteristics of the fractions were as follows:

- F1: for particles smaller than or equal to 500 μm;
- F2: for particles smaller than 1 mm and larger than 500 μm;
- F3: for particles larger than 1 mm.

Digestion tests were carried out according to the 3051A EPA method, in triplicate, for F1, F2, and F3 to quantify the contents of the metals which compose them and thus decide which fraction of interest was to be studied.

### 2.3.2. Heat Pretreatment

To carry out the thermal pretreatment, 100 g of the sample of the fraction of interest was placed in porcelain crucibles, which were kept in an oven at a temperature of 650 °C and ambient atmosphere; this procedure was carried out for 1 h and 3 h. After the thermal treatment, the samples were allowed to cool down at room temperature. In the next step of the process, they were submitted to leaching with sulfuric acid and DL-malic acid under the same conditions used in Section 2.3.3. For comparison purposes, samples without heat pretreatment were sent directly to the leaching stage.

### 2.3.3. Sulfuric Acid and DL-Malic Acid Leaching

For the leaching of the fraction of interest, with and without heat pretreatment, some optimized conditions in previous studies were adopted for temperature, time, and solid/liquid ratio [25] (room temperature, 2 h, $R_{S/L}$: 75 g/L). The experiments were carried out with 1.5 M DL-malic acid and 2 M sulfuric acid [26,27] under constant agitation. For each acid, tests were performed with and without the addition of the oxidizing agent $H_2O_2$ 10% *v/v* [25]. After leaching, the samples—all of them in triplicate—were filtered and swelled to 100 mL; then, an aliquot was taken for element quantification via ICP-OES.

The nomenclature used (A, B, C, ... ) in the experiments to denote varying types of leaching agent, the addition or not of the oxidizing agent, and the lack of heat pretreatment and presence of heat pretreatment for different lengths of time, is presented in Table 1.

**Table 1.** The nomenclature used in different leaching experiments.

| Sample | Acid | Oxidant Agent | Heat Pretreatment |
|---|---|---|---|
| A | Sulfuric 2 M | - | - |
| B | Sulfuric 2 M | $H_2O_2$ 10% *v/v* | - |
| C | DL-Malic 1.5 M | - | - |
| D | DL-Malic 1.5 M | $H_2O_2$ 10% *v/v* | - |
| E | Sulfuric 2 M | - | 1 h |
| F | Sulfuric 2 M | $H_2O_2$ 10% *v/v* | 1 h |
| G | Sulfuric 2 M | - | 3 h |
| H | Sulfuric 2 M | $H_2O_2$ 10% *v/v* | 3 h |
| I | DL-Malic 1.5 M | - | 1 h |
| J | DL-Malic 1.5 M | $H_2O_2$ 10% *v/v* | 1 h |
| K | DL-Malic 1.5 M | - | 3 h |
| L | DL-Malic 1.5 M | $H_2O_2$ 10% *v/v* | 3 h |

## 3. Results and Discussion

### 3.1. LIB Collection

For the batteries that are interesting for the development of the work, 404 units of different brands were selected, as shown in Table 2, and the others were returned to the technical assistance collection system.

**Table 2.** Batteries are sorted and selected for the work development.

| Brand | Quantity | Percentage (%) |
|---|---|---|
| Samsung | 178 | 44.0 |
| Nokia | 49 | 12.1 |
| LG | 28 | 6.9 |
| Motorola | 37 | 9.2 |
| Apple | 19 | 4.7 |
| Others | 93 | 23.0 |
| Total | 404 | 100.0 |

*3.2. Characterization*

The five most common models/brands of LIBs that were chosen to perform the characterization are shown in Figure 1a,b shows one of them, after manual disassembly, with each part that composes it.

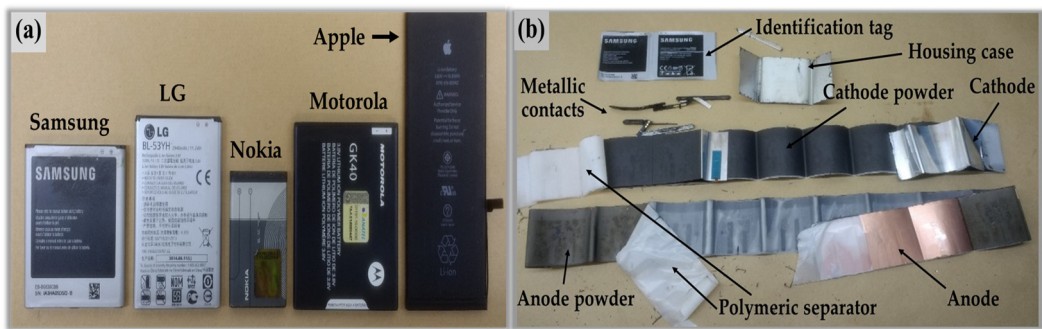

**Figure 1.** The five models/brands chosen (**a**); and one of them disassembled (**b**).

The XRF analysis of the housing case and cathode and anode collectors is presented in Table 3. By analyzing the data presented, it is verified that the majority composition is more than 99% Al and almost 100% Cu for the cathode and anode collectors, respectively. However, in a slightly lower percentage (all samples with a content >95%), the housing case is composed of aluminum alloys.

**Table 3.** Housing case and collector compositions via XRF analysis.

| Sample | Housing Case | | Cathode Foil | | Anode Foil | |
|---|---|---|---|---|---|---|
| | Al (%) | Others (%) | Al (%) | Others (%) | Cu (%) | Others (%) |
| Samsung | 96.0 | 4.0 | 99.4 | 0.6 | 99.9 | 0.1 |
| LG | 96.5 | 3.5 | 99.1 | 0.9 | 99.9 | 0.1 |
| Nokia | 97.6 | 2.4 | 99.0 | 1.0 | 99.9 | 0.1 |
| Motorola | 95.0 | 5.0 | 99.6 | 0.4 | 99.8 | 0.2 |
| Apple | 95.8 | 4.2 | 99.8 | 0.2 | 99.3 | 0.7 |

All analyses carried out for the cathode powder are presented below (XRF, XRD, SEM, and ICP-OES). As shown in Table 4, the majority of the cathode powder composition of LIBs consists of cobalt, oxygen, and fluorine, but in this analysis, the percentage of lithium was not accounted for, as the XRF technique is not able to detect it due to its low atomic weight. The fluorine presence is due to the PVDF composition, a polymer used as a binder for the cathode powder on the support foil (cathode collector).

**Table 4.** Cathode powder XRF analysis.

| Sample | Element | | | | |
|---|---|---|---|---|---|
| | Co (%) | O (%) | F (%) | Al (%) | Others (%) |
| Samsung | 62.6 | 25.5 | 9.3 | 0.6 | 2.0 |
| LG | 65.2 | 26.6 | 6.8 | 0.4 | 0.9 |
| Nokia | 62.5 | 25.5 | 11.3 | 0.4 | 0.4 |
| Motorola | 63.0 | 25.7 | 8.4 | 1.0 | 1.9 |
| Apple | 64.9 | 26.5 | 7.1 | 0.9 | 0.7 |

According to XRD analysis, as shown in Figure 2, all samples essentially show the peaks of the LiCoO2 phase, as indexed by the crystallographic chart 00-016-0427. However, some slightly broadened and low-intensity peaks suggest the presence of the $CoCo_2O_4$ phase.

In this regard, despite the possibility that the $CoCo_2O_4$ phase is present, the diffraction peaks were indexed by the $LiCoO_2$ phase, as the broadening of certain peaks may originate from a disorder of planes (hkl) at the crystalline structure level, among other possible reasons.

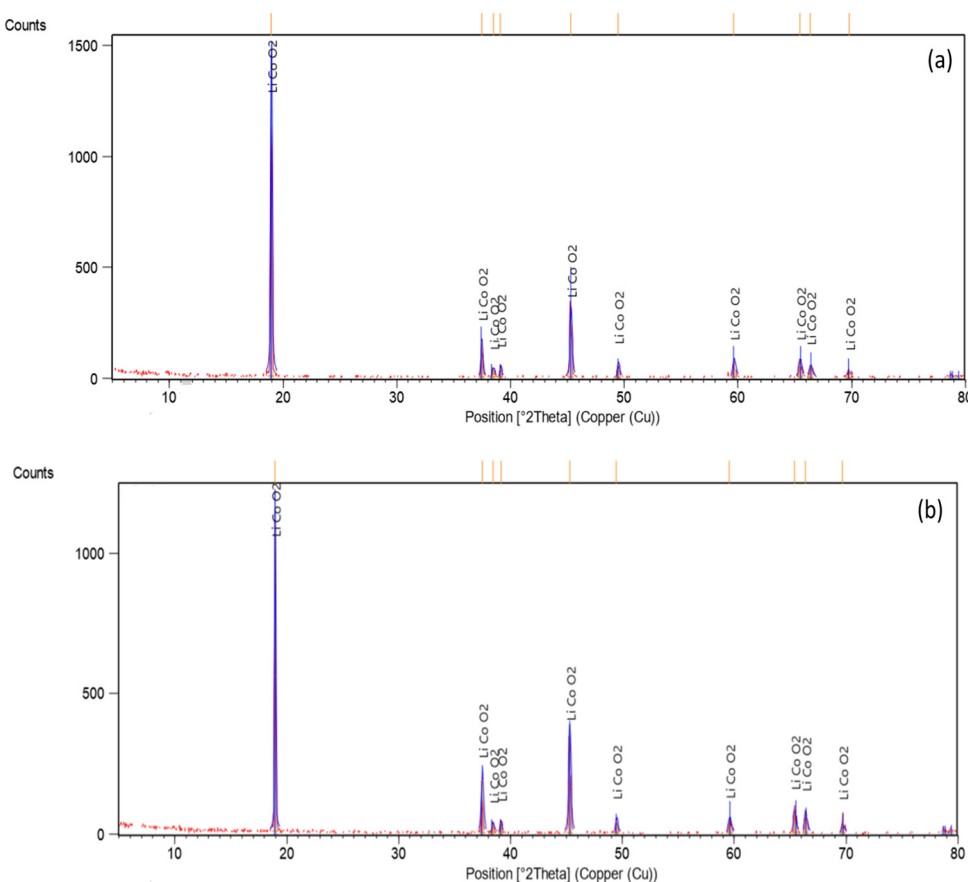

**Figure 2.** *Cont.*

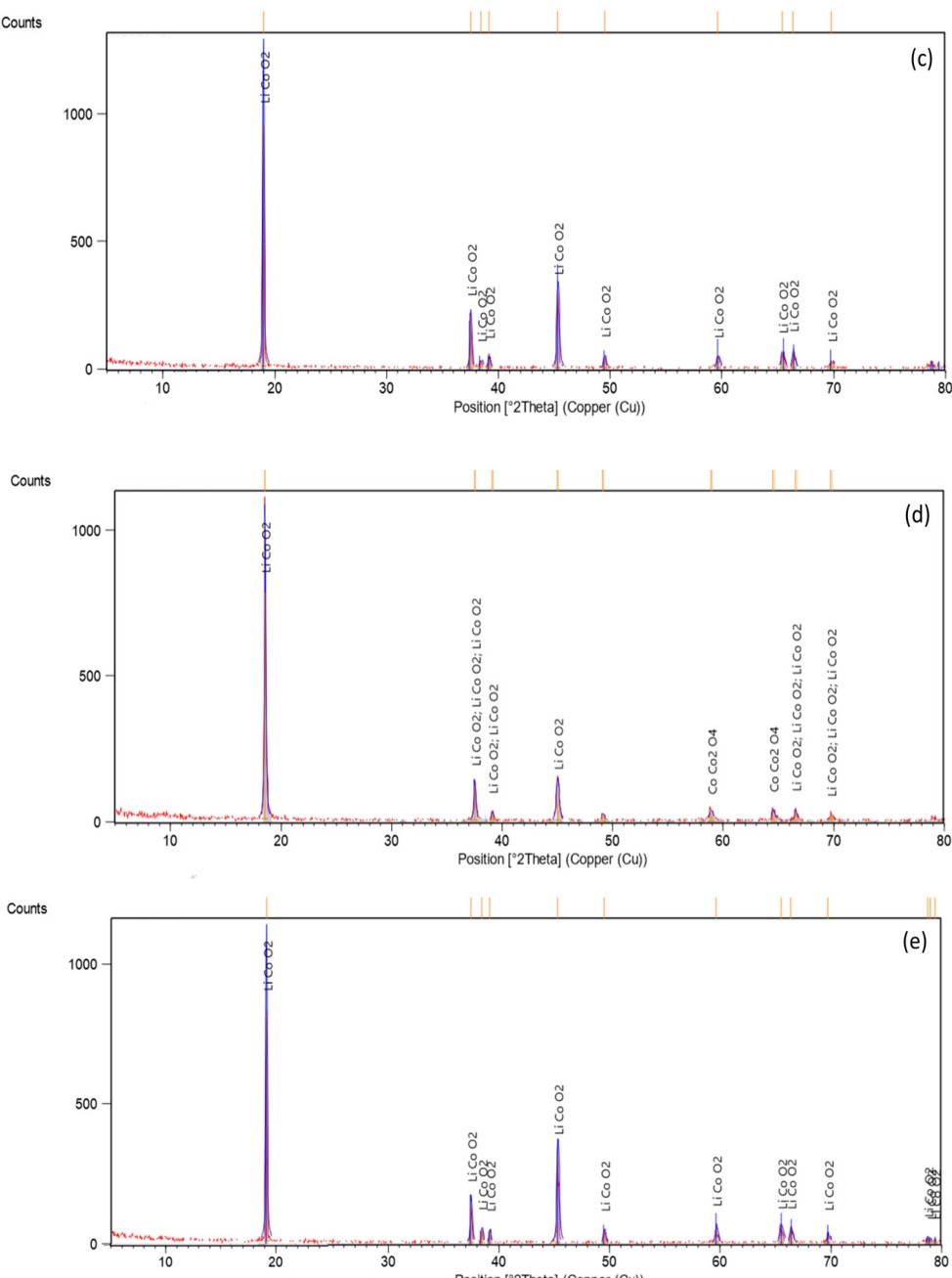

**Figure 2.** XRD analysis of cathode powder: Samsung (**a**), LG (**b**), Nokia (**c**), Motorola (**d**), and Apple (**e**).

All samples analyzed via SEM were subjected to the same voltage (15 kV), and all the images shown in Figure 3 have the same zoom (1000). Regarding the scale, for all images in the left bottom corner, the length is 80 μm. Then, when evaluating them, it can be seen that the cathode powder grain size is similar for all samples (a, b, d, and e), except for Nokia (c), which presented a smaller grain size.

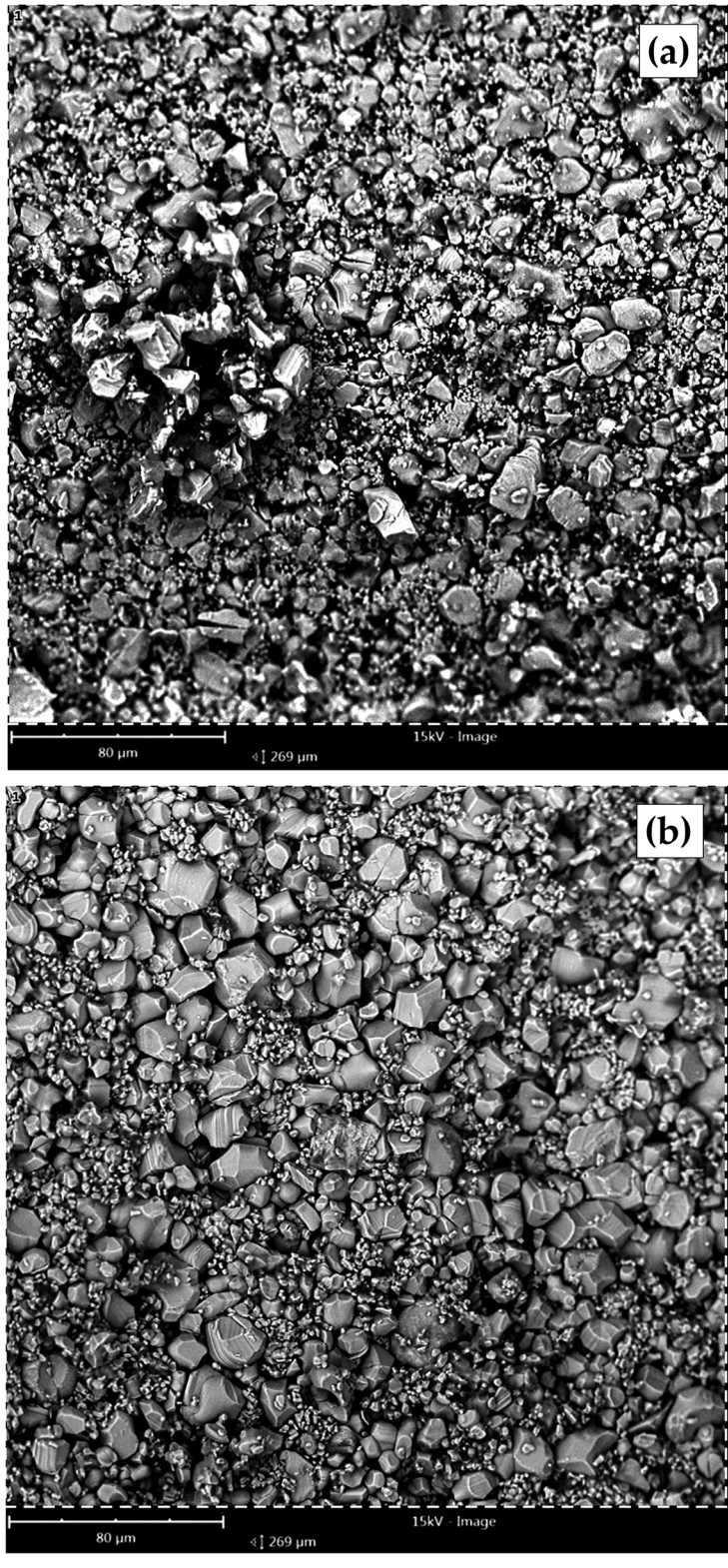

**Figure 3.** *Cont.*

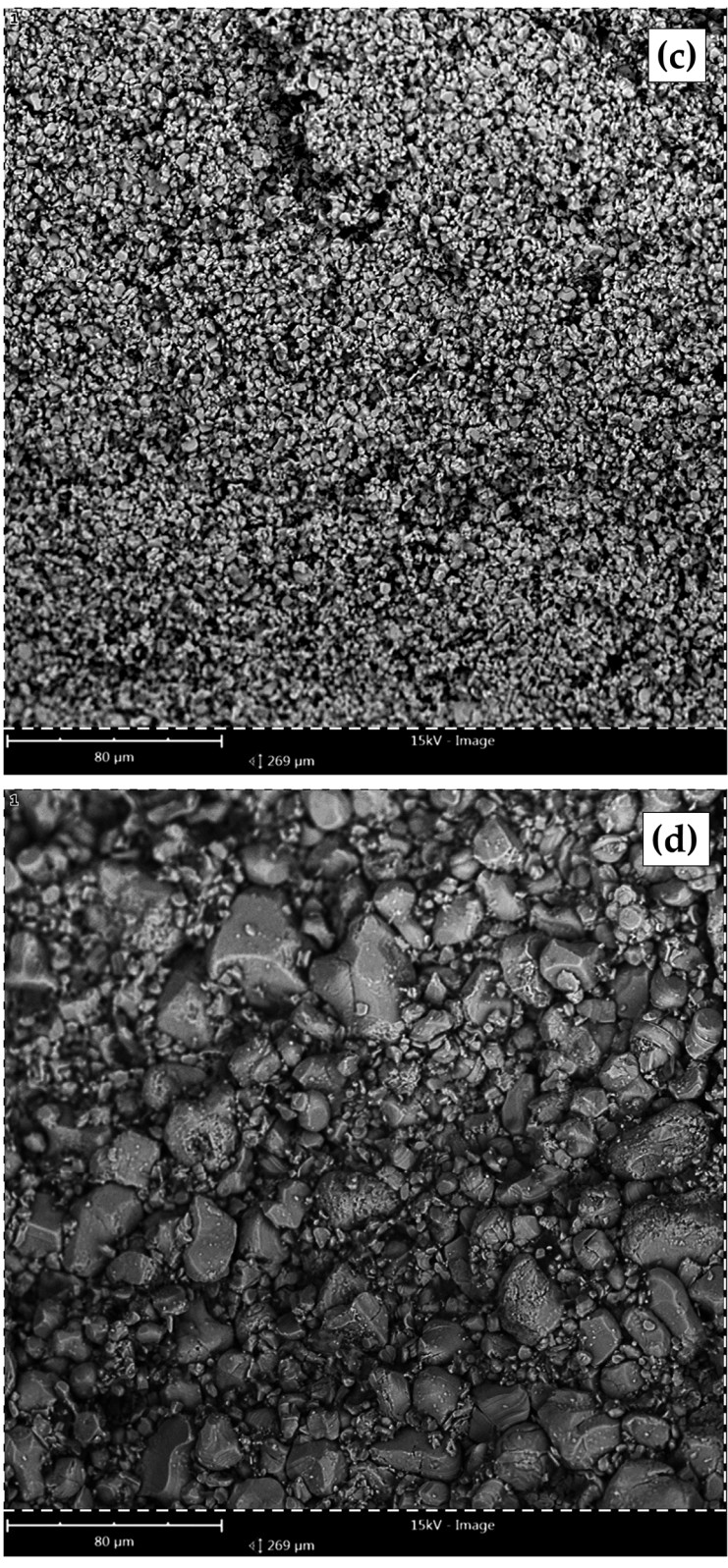

**Figure 3.** *Cont.*

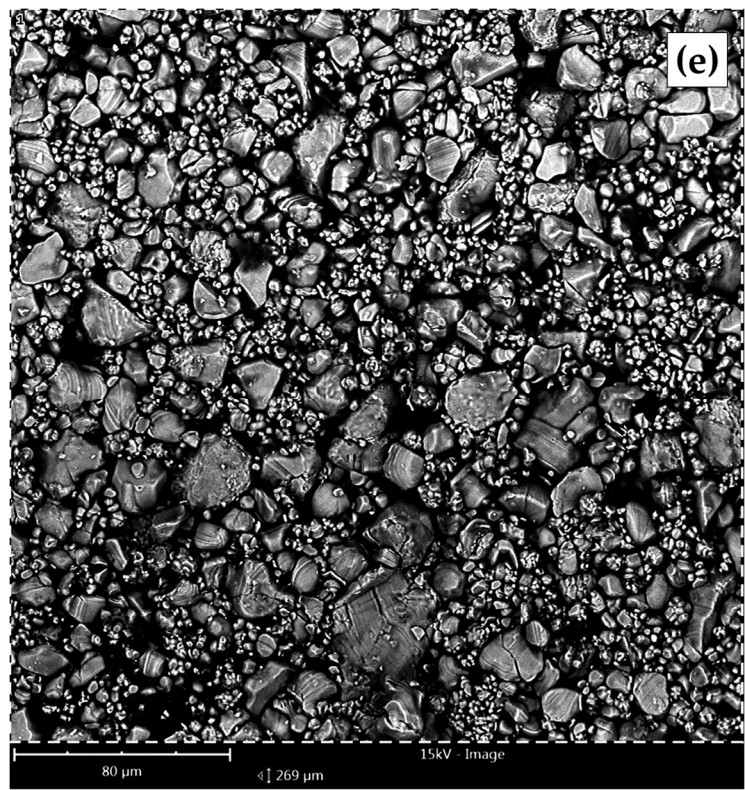

**Figure 3.** Grain size detail of LIBs' cathode powders by SEM (15 kV, scale bar = 80 μm, 1000 magnification) for Samsung (**a**), LG (**b**), Nokia (**c**), Motorola (**d**), and Apple (**e**).

Using the previously mentioned techniques, it was possible to identify the elements which composed the LIBs. ICP-OES analysis was used, as it is more sensitive to quantify the composition. When assessing the data presented in Table 5, it is verified that, for the five LIBs, the majority of the cathode powder composition comprises cobalt and lithium; the aluminum presence is due to removing some parts from the cathode foil during scraping process. The percentage remaining to complete the total initial mass added in the leaching process is due to possible amounts of Al, Co, Li, and Mn not leached; oxygen and fluoride are also present in this material, though they cannot be detected by the technique used.

**Table 5.** Chemical analysis of cathode powder via ICP-OES.

| Sample | Element | Weight (%) | SD (%) |
|--------|---------|------------|--------|
| Samsung | Al | 0.10 | 0.01 |
| | Co | 58.00 | 1.69 |
| | Li | 6.15 | 0.10 |
| | Mn | 0.01 | 0.00 |
| LG | Al | 0.04 | 0.01 |
| | Co | 52.35 | 0.82 |
| | Li | 5.40 | 0.27 |
| | Mn | 0.00 | 0.00 |
| Nokia | Al | 0.02 | 0.01 |
| | Co | 53.41 | 0.61 |
| | Li | 4.99 | 0.02 |
| | Mn | 0.00 | 0.00 |

**Table 5.** *Cont.*

| Sample | Element | Weight (%) | SD (%) |
|---|---|---|---|
| Motorola | Al | 0.24 | 0.03 |
| | Co | 60,96 | 1.95 |
| | Li | 6.43 | 0.04 |
| | Mn | 0.00 | 0.00 |
| Apple | Al | 0.01 | 0.01 |
| | Co | 52.60 | 3.01 |
| | Li | 5.76 | 0.05 |
| | Mn | 0.00 | 0.00 |

The results regarding the contents from the different analyses performed—XRF, XRD, SEM, and ICP-OES—support LIBs as a secondary source of lithium and cobalt [26,28].

To carry out the cathode powder heat pretreatment, it was necessary to know the degradation temperature of the PVDF binder. For this reason, a thermogravimetric analysis of polyvinylidene fluoride was performed with a heating ramp of 20 °C/min in the presence of atmospheric air. Judging from the TGA results, shown in Figure 4a, it turns out (green curve) that, approximately, at a temperature of 630 °C, the mass loss was complete (similar to Kim et al.) [29]; therefore, the pretreatment experiment to remove PVDF from the F1 fraction was carried out at 650 °C to ensure the effectiveness of the process.

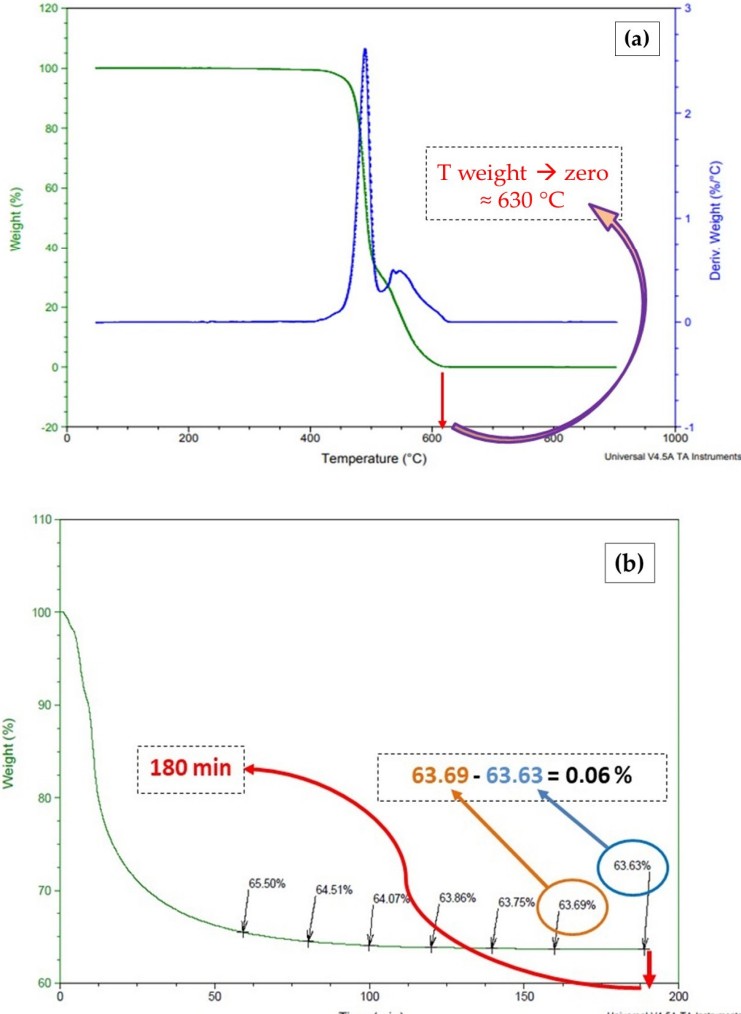

**Figure 4.** Thermal degradation curve for pure PVDF (**a**) and heat pretreatment time to remove PVDF from F1 fraction (**b**).

The remaining mass after 160 min of heat treatment was 63.69% of the original value, and after 180 min it was 63.63%—as shown in Figure 4b—which justifies ceasing the test, as this percentage difference (0.06%) is lower than equipment detection error limit (0.10%), corroborating the results of Natarajan et al. [30].

### 3.3. Hybrid Processing

#### 3.3.1. Comminution and Granulometric Separation

Before comminution, all 399 LIBs were weighed, totaling 13,068.6 g; after comminution, the total weight was 12,090.7 g, equating to a mass loss of 977.9 g (7.48%) due to gas evaporation and very light particles being dragged through the exhaust system. The total mass was divided by sieve separation, resulting in three fractions (F1, F2, and F3) which are shown in Figure 5. In the background fraction (F1) were particles smaller than 500 μm (a), in the intermediate sieve (F2) were particles smaller than 1 mm and larger than 500 μm (b), and in the upper sieve (F3) were particles larger than 1 mm (c). The percentages of the total mass for each fraction are 69.1% (F1), 12.8% (F2), and 18.1% (F3).

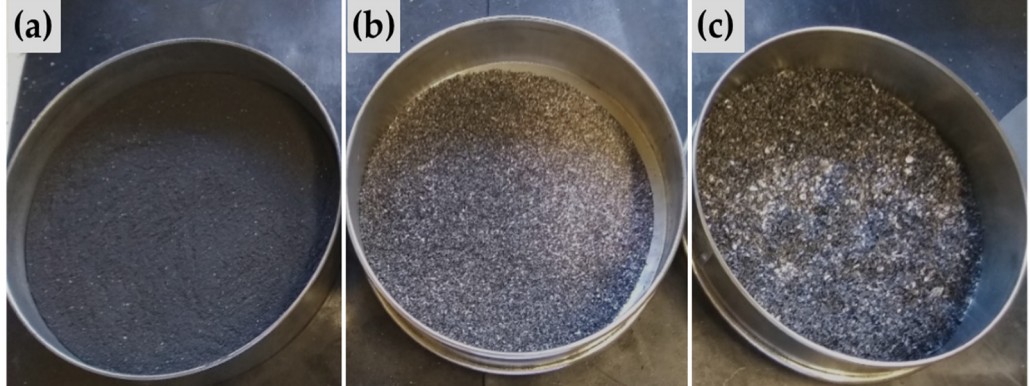

**Figure 5.** Sample fractions after the comminution process and granulometric separation. F1: under 500 μm (**a**); F2: 500–1000 μm (**b**); F3: above 1000 μm (**c**).

The contents of each analyzed element in the three fractions (F1, F2, and F3), are presented in Table 6. The particles smaller than 500 μm represent the highest mass percentage; this is the fraction where the material of interest ($LiCoO_2$) to be recovered was found.

**Table 6.** Mass percentage of elements that make up the fractions.

| Element | F1 | | F2 | | F3 | |
|---|---|---|---|---|---|---|
| | Weight (%) | SD (%) | Weight (%) | SD (%) | Weight (%) | SD (%) |
| Al | 6.90 | 0.56 | 6.49 | 0.18 | 10.34 | 0.18 |
| Co | 47.87 | 4.66 | 11.21 | 0.16 | 5.40 | 0.30 |
| Cu | 5.24 | 0.33 | 54.22 | 2.01 | 17.58 | 2.57 |
| Fe | 0.48 | 0.01 | 1.13 | 0.10 | 1.33 | 0.25 |
| Li | 6.49 | 0.65 | 1.56 | 0.04 | 0.72 | 0.03 |
| Mn | 1.49 | 0.09 | 1.66 | 0.02 | 1.79 | 0.12 |

Due to F1 containing the highest mass percentage of elements of interest to be recovered, 47.87% for cobalt and 6.49% for lithium, it was chosen as the fraction on which to focus in the work.

#### 3.3.2. Leaching Process and Heat Pretreatment

- Oxidizing agent influence:

The percentages of elements extracted after sample leaching performed without heat pretreatment by the use of 2 M sulfuric acid (A and B) and 1.5 M DL-malic acid (C and D)

are presented in Table 7. In experiments B and D, the oxidizing agent $H_2O_2$ was added, and an improvement in leachate content was observed. For the metals of interest—cobalt, and lithium—leached by sulfuric acid, the efficiencies were improved by 26.72% and 9.24%, respectively. When leaching was performed with DL-malic acid, the improvement was more significant, resulting in gains of 57.71% for cobalt and 32.80% for lithium. The rest of the mass composing the total is made up of Al, Co, Cu, Fe, Li, and Mn, which have not been leached, and $O_2$ and graphite, which are not leachable.

In the samples leached by sulfuric acid and submitted to heat pretreatment for 1 h (E and F), according to Table 7, a recovery percentage from 28.65 to 33.49% for Co was observed when the oxidizing agent was added. For lithium, the content increased from 4.10 to 4.63% after adding $H_2O_2$. When the samples were subjected to 3 h of heat pretreatment (G and H), the contents for the best condition (with the oxidizing agent) were 36.36% for Co and 4.64% for Li.

For those samples leached by DL-malic acid and subjected to 1 h of heat pretreatment (I and J), the best contents were also obtained with the addition of the oxidizing agent. Improvements from 22.25 to 29.78% for Co and from 2.45 to 3.44% for Li were observed. After 3 h of heat pretreatment, the same behavior was observed, and the recovery improved from 22.81 to 32.73% for Co and from 3.55 to 3.99% for Li.

**Table 7.** Leaching percentages by sulfuric acid and DL-malic acid for F1 fraction.

| Sample | Al (%) | | Co (%) | | Cu (%) | | Fe (%) | | Li (%) | | Mn (%) | |
|---|---|---|---|---|---|---|---|---|---|---|---|---|
| | Weight | SD | Weight | SD | Weight | SD | Weight | SD | Weight | SD | Weight | SD |
| A | 2.39 | 0.02 | 20.58 | 0.60 | 0.16 | 0.03 | 0.1 | 0.02 | 2.92 | 0.02 | 1.83 | 0.05 |
| B | 2.14 | 0.09 | 26.08 | 0.46 | 2.44 | 0.19 | 0.07 | 0.01 | 3.19 | 0.05 | 3.07 | 0.26 |
| C | 0.87 | 0.03 | 11.54 | 0.23 | 0.07 | 0.01 | 0.03 | 0.01 | 1.89 | 0.08 | 2.08 | 0.06 |
| D | 1.20 | 0.03 | 18.29 | 0.37 | 2.74 | 0.23 | 0.03 | 0.01 | 2.51 | 0.20 | 2.35 | 0.11 |
| E | 2.59 | 0.02 | 28.65 | 0.24 | 0.00 | 0.00 | 0.22 | 0.00 | 4.1 | 0.04 | 2.96 | 0.02 |
| F | 2.61 | 0.12 | 33.49 | 0.44 | 2.64 | 0.38 | 0.14 | 0.01 | 4.63 | 0.07 | 2.91 | 0.06 |
| G | 2.54 | 0.07 | 32.40 | 0.49 | 0.00 | 0.00 | 0.23 | 0.03 | 4.74 | 0.33 | 3.47 | 0.10 |
| H | 2.42 | 0.15 | 36.36 | 0.57 | 2.61 | 0.70 | 0.22 | 0.01 | 4.64 | 0.21 | 3.52 | 0.06 |
| I | 0.47 | 0.01 | 22.25 | 0.11 | 0.00 | 0,00 | 0.07 | 0.00 | 2.45 | 0.06 | 2.75 | 0.03 |
| J | 0.78 | 0.04 | 29.78 | 0.58 | 2.12 | 0.03 | 1.24 | 0.01 | 3.44 | 0.41 | 2.88 | 0.08 |
| K | 0.63 | 0.05 | 22.81 | 0.83 | 0.00 | 0.00 | 0.12 | 0.02 | 3.55 | 0.20 | 3.38 | 0.25 |
| L | 0.81 | 0.04 | 32.73 | 0.52 | 1.88 | 0.05 | 0.09 | 0.01 | 3.99 | 0.21 | 3.25 | 0.06 |

- Pretreatment influence:

In order to better comprehend the meaning of the results and to allow for easier visual data understanding, a graphical representation was built, as shown in Figure 6; however it was made only for the elements of interest, cobalt and lithium.

According to Figure 6a, when comparing samples B, F, and H (in the case of cobalt leached by sulfuric acid and $H_2O_2$), an increase was observed from B to F (26.08 to 33.49%, gain of 28.41%), and a smaller improvement was seen from F to H (33.49 to 36.36%, gain of 8.56%). Performing this analysis for samples D, J, and L (in the case of cobalt leached by DL-malic acid and $H_2O_2$), a great improvement was observed from D to J (18.29 to 29.78%, gain of 62.82%), and a gain of 9.90% (29.78 to 32.73%) was seen from J to L. In both situations a larger improvement occurred when switching from non-pretreatment to 1 h of pretreatment than from 1 h to 3 h of pretreatment.

As displayed in Figure 6b, an improvement was observed (for lithium leached by sulfuric acid and $H_2O_2$) from sample B to F (3.19 to 4.63%, gain of 45.14%), but no improvement was seen from F to H. When the samples were leached by DL-malic acid and $H_2O_2$, a great increase was observed from D to J (2.51 to 3.44%, gain of 37.05%); however a smaller improvement was observed from J to L (3.44 to 3.99%, gain of 15.98%).

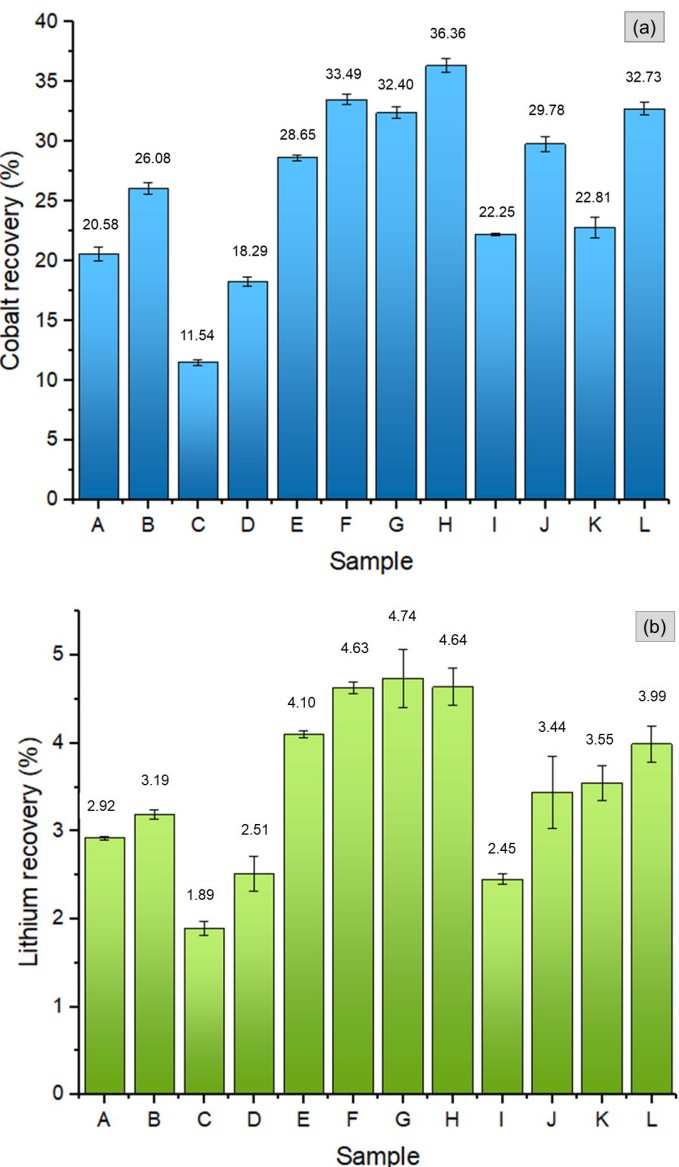

**Figure 6.** Percentage recovery according to the sample processing for cobalt (**a**), and for lithium (**b**).

Evaluating the elemental contents of the leaching liquors with sulfuric acid for sample F (1 h of pretreatment and with $H_2O_2$) and for sample H (3 h of pretreatment and with $H_2O_2$), due to a lengthier time of heating, a gain of 8.56% (increasing from 33.49 to 36.36%) was observed for cobalt, and no gain was observed for lithium. Performing this same analysis for DL-malic acid, in sample J (1 h of pretreatment and with $H_2O_2$) and sample L (3 h of pretreatment and with $H_2O_2$), there was a gain of 9.90% (from 29.78 to 32.73%) for cobalt, and the improvement for lithium was within the standard deviation range. However, an economic feasibility study is necessary to determine whether the comparison between the energy cost spent for an additional 2 h of heat pretreatment and the gain in recovery efficiency will be positive or negative.

From an environmental viewpoint, the use of DL-malic acid as a leaching agent is feasible when it is compared to sulfuric acid, which is widely used in established hydrometallurgical processes [31] and causes more damage to the environment. According to Equation (1), DL-malic acid reached 88.92%, and 90.02% in cobalt recovery efficiency for

1 h and 3 h of heat pretreatment, respectively; in the case of lithium, the recovery efficiency reached 74.30% (1 h) and 85.99% (3 h).

$$E_{Dl-Malic f(t)} = \frac{R_{DL-Malic f(t)}}{R_{Sulfuric f(t)}} \times 100 \qquad (1)$$

where the terms are defined as follows:

- $E_{DL-Malic\,f(t)}$: DL-malic acid recovery efficiency based on sulfuric acid recovery efficiency, as a function of time, as a percentage (%);
- $R_{DL-Malic\,f(t)}$: DL-malic acid recovery efficiency as a function of time, as a percentage (%);
- $R_{Sulfuric\,f(t)}$: Sulfuric acid recovery efficiency as a function of time, as a percentage (%).

In the following steps, the impurities (Al, Fe, and Mn) must be removed, e.g., by precipitation [32,33]; Cu and Co [25,33,34] can be obtained by selective extraction, and Li can be obtained as a salt compound by liquid evaporation because it will be the last one present in the leaching liquor.

Taking samples F and H—leached by sulfuric acid—and J and L—leached by DL-malic acid—as the best leaching conditions, it is possible to extrapolate a recovery value per ton of LIBs processed, considering a total recovery of the elements just for comparison purposes. For the process carried out with sulfuric acid, 334.9 kg (1 h) and 363.6 kg (3 h) of Co and 46.3 kg (1 h), and 46.4 kg (3 h) of Li could be recovered per ton of LIBs. If DL-malic acid were used, it would be possible to obtain 297.8 kg (1 h) and 327.3 kg (3 h) of Co and 34.4 kg (1 h) and 39.9 kg (3 h) of Li per ton of LIBs. The most abundant cobalt ores have 355.2 kg, 295.3 kg, and 179.5 kg of Co per ton of cobaltite, erythrite, and skutterudite, respectively [35,36]. The fraction of interest evaluated in this study (F1), from obsolete LIBs, has the potential to be a secondary source of cobalt when compared to metal ores' content. The most exploited lithium ore is spodumene (70 kg per ton) [37] and, although the percentage recovered from LIBs is about half of that found in ore, they can still be an important secondary source of the metal [15].

## 4. Conclusions

The battery characterization showed data consistent with the literature; the supporting foils of the cathode and anode are mostly (more than 95%) composed of aluminum and copper, respectively. It was also verified that the fraction with the highest content of the metals of interest (Co and Li) had particles smaller than 500 μm in size and represented 69.1% of the total mass comminuted.

An important conclusion was obtained from the heat pretreatment study. The total decomposition (when the mass tends toward 0) of PVDF, which is present as a cathodic powder binder, occurs around 630 °C, and for samples that were subjected to heat pretreatment, the best results were obtained for 1 and 3 h of processing. However an economical evaluation is necessary to determine if the cost of 2 h more of heat processing can be justified by the efficiency improvement.

In addition to the heat pretreatment, the main objective of this work was achieved by obtaining a more environmentally friendly metallurgical route through the use of an organic leaching agent. By comparing the best conditions for the two acids used in this process, it was found that DL-malic acid reached a leaching potential of 88.92% for Co and 74.30% for Li compared to sulfuric acid's leaching potential (inorganic and not environmentally friendly). The use of malic acid, associated with an oxidizing agent and heat pretreatment for 1 h and 3 h, proved to be promising, with extraction contents very close to those of sulfuric acid.

**Author Contributions:** Conceptualization, H.M.V. and J.C.M.M.; methodology, H.M.V. and J.C.M.M.; formal analysis, J.C.M.M.; investigation, J.C.M.M.; resources, J.C.M.M.; data curation, H.M.V. and J.C.M.M.; writing—original draft preparation, J.C.M.M.; writing—review and editing, H.M.V. and J.C.M.M.; supervision, H.M.V.; project administration, H.M.V. All authors have read and agreed to the published version of the manuscript.

**Funding:** This research received no external funding.

**Data Availability Statement:** Not applicable.

**Acknowledgments:** The authors are grateful to Capes, CNPq, FINEP, and FAPERGS (Brazil) for their financial support.

**Conflicts of Interest:** The authors declare no conflict of interest.

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
