# Peer review of "Development of a More Sustainable Hybrid Process for Lithium and Cobalt Recovery from Lithium-Ion Batteries"

_minerals, doi:10.3390/min13060798_

Round 1
Reviewer 1 Report
This study investigated the recovery of Li and Co from lithium-ion batteries by combination of heat treatment and leaching process. However, this manuscript was not well written many mistakes were found. To improve the quality of this manuscript, please carefully consider the comments and suggestions below.
Abstract:
The abstract of this manuscript was not well written. Too much background. Moreover, the research methodology is not clearly present that leading the reader difficult to understand on the results and finding. For more attractive to the readers, the authors should revise this abstract and add more significant content as well as method, results/finding, and discussion/conclusion.
1. Introduction:
The overall content in the introduction was workable. However, the authors should carefully clarify the pretreatment process and separation process as well (line 66-72). This paragraph might make a confused with outfield reader when they read.
2. Materials and Methods:
Figure 1: Please identify the sub-figure caption in figure 1 by adding (a) and (b) in the figure.
Line 190: Please recheck abbreviation RXD, it must be XRD.
Please check all abbreviations in manuscript, it should be note that all abbreviation must present the full word in the first time of presenting in main text. In addition, some abbreviations are wrong representatives (for example “FRX” in line 125, 129, 183, 188, 190, 193, 232). It must be XRF rather than FRX.
3. Results and Discussion:
This section was not well managed following the research methodology of this work. The author should revise and restructure this section. In addition, the comparison of the process with and without heat treatment is not clearly present.
Figure 5: Please revise the figure caption, this caption is difficult to understand especially for size fraction. The authors should present the size fraction in common form as academic writing.
Table 7 and 8 should be combined as one table.
Line 270: Please check “Table 7”, it should be removed.
4. Conclusion
This section number must be number 4 not 5.
Reference
Please kindly check reference format in this manuscript, there are many wrong format were detected as well as authors name.
In addition, please kindly check for many mistakes in this manuscript as listed below.
- There are many wrong typos detected in the manuscript, for example “poder” in line 225. The authors should carefully check for the final version before submitting to the journal.
- Line 77: Please recheck unknown comma.
- Line 129: Please check latter “e”. What does it mean?
- Line 174: Please remove duplicate word “Table 2”.
- Figure 2: Please revise all sub-figures in figure 2 due to some grey color appearing on “2Theta” axle.
- Figure 3: Please revise figure alignment, all figures overlap.
- Table 7, 8: Please remove grey background in table.
- Text citation: Please check the whole citation format in the main text of this manuscript follow by guide for the author that provide on journal homepage.
Please carefully check the misspelling.
Reviewer 2 Report
Apart from minor language polishing that the manuscript needs (mainly in abstract), I could go through it fast and easy.
I see a lot of effort has been done in material preparation, realization of the tests and formulating the results and characterizations into a manuscript. Yet I like to give my comments in hopes of turning this work into a better version and this mat happen after a major revision.
1- There are parts like "The useful life is about 2 years or 500 cycles ..." with no references provided. While I do not mean to provide references in the abstract, but for delivering such data some level of precision is required in the text.
2- The results mentioned in the abstract do not seem valid. The same is repeated in the conclusion, where "The best yields occurred after a heat pretreatment of 1 h". Based on Table 8, the best recoveries are in the case of 3h heat treatment and use of peroxide. (I understand the point where authors like to mention PROBABLY the economy of 2 hours longer heat treatment is not favorable, yet no data is provided in this regard even for the case of 1h treatment or the $ gain due to increase of recovery).
3- X-ray characterization methods are generally mentioned as XRD and XRF in most of the scientific literature.
4- Figure 2 can be merged into one figure with multiple lines for better comparison if necessary.
5- I could not relate how " leaching potential of 88.92% for Co and 74.30% for Li" are obtained. The methodology needs more clarification.
6- I have difficulty in understanding the data provided in table 7 and 8. How the reader should interpret these numbers? A graphical representation of the total recoveries (based on the metals transferred into leachate over the total metal present in the solid) would be more meaningful and easy to follow by the readership.
7- I like the fact that the leaching has been performed at room temperature, however, with no kinetics data it is hard to justify if the final recoveries are feasible. Maybe increasing the leaching time to 4 hours help in recovering more metals and maybe 60 min is already enough!
8- there is no section on how to separate and recover the leached metals from the leachate.
Minor works are needed throughout the text to make it more readable.
Round 2
Reviewer 1 Report
This manuscript was revised as per reviewer comment. However, there are some mistakes that are detected as listed below.
Abstract:
The whole content and structure in this abstract were improved compared with the previous version. However, there are some abbreviations that were presented without any declaration of the full word (i.e., DL-malic, PVDF). The authors should revise this point to increase a wide range of readers especially for outfield reader.
There are some fluctuations of synonyms used in this manuscript, the authors should use only one pattern for avoid the difficulty of understanding when reading; For example, “DL-malic”, “Dl-malic”, “C4H6O5”—The authors should use only one of these synonyms in main text of manuscript.
There are some fluctuations used between comma and dot for mathematical sign, for example in line 299 “8,56 must be 8.56”; line 302 “9,90 must be 9.90”.
Fig.3: All the scale bars are difficult to read due to their low resolution. The authors should revise these photos or specify the magnification of these SEM images.
Fig.5: Identification of size range must be: F1: –500 µm (a); F2: 500–1000 µm (b); F3: +1000 µm (c)
Equation 1: Please check the definition of variable in equation 1 (i.e., RDl-sulfuric f(t) must be Rsulfuric f(t))
References:
The reference list is still in the wrong format, please kindly check list number 1, 19, 20, 26, 28, 30, 35, 36, 38. In addition, the authors should read the guide for authors to clearly understand in reference format.
Reviewer 2 Report
The revised version seems much better to me. I am happy to see the improvements.
The manuscript is more readable now. I have no specific point to mention.
